# Comparative Study on the Scratch and Wear Resistance of Diamond-like Carbon (DLC) Coatings Deposited on X42Cr13 Steel of Different Surface Conditions

Shiraz Ahmed Siddiqui * and Maria Berkes Maros 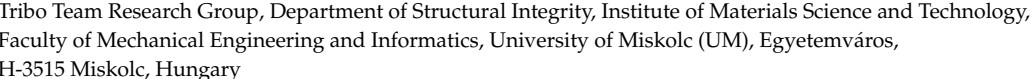

Tribo Team Research Group, Department of Structural Integrity, Institute of Materials Science and Technology, Faculty of Mechanical Engineering and Informatics, University of Miskolc (UM), Egyetemváros, H-3515 Miskolc, Hungary
* Correspondence: ahmedshiraz432@gmail.com

**Abstract:** Tribological investigations are of great importance, especially in the case of novel combinations of materials used for the tribosystem. In the current research, multilayer diamond-like carbon coating deposited by plasma enhanced chemical vapour deposition on an X42Cr13 plastic mould tool steel is studied with two different surface conditions of the substrate. On the one hand, it is secondary hardened; on the other hand, it is additively plasma nitrided preceding the diamond-like carbon coating. This latter combined treatment, called duplex treatment, has an increasingly wide range of applications today. However, its effectiveness largely depends on applying the appropriate nitriding technology. The tribological behaviour was characterised by an instrumented scratch test and a reciprocating ball-on-plate wear test. The results demonstrate better scratch resistance for the duplex-treated samples, while they show weaker performance in the applied wear type of loading. The current comparative study reveals the reason for the unexpected behaviour and highlights some critical aspects of the heat treatment procedure. The architecture of the tested multilayer DLC coating is unique, and no tribological results have yet been published on tribosystems combined with an X42Cr13 steel substrate. The presented results may particularly interest tribologists and the materials research community.

**Keywords:** multilayer DLC coating; CrN interlayer; X42Cr13 tool steel; duplex treatment; nitriding; scratch test; reciprocating wear test; damage mechanism





## 1. Introduction

Duplex surface treatment is a unique method combining two (or more) surface technologies to produce a composite surface having improved properties [1]. The significant efforts and advances in this field have led to considerable developments in reducing the wear, friction, and damage of the components, thereby increasing the durability of the components [2–5]. The duplex treatment combines surface modification and surface coating [6–8]. Surface modification refers to the changes in the microstructure, realised by thermo-chemical diffusional processes (such as nitriding, carburising), mechanical processes (such as forging, shot blasting, etc.), or ion implantation. Surface coating is a technique of adding a new layer onto the underlying substrate. The coating can be produced from the vapour state of the constituents, chemically (sol-gel, anodising, etc.) or by thermal spray (air plasma, vacuum plasma). The vapour state method is most widely used in mechanical engineering industries. One of the main objectives of surface coating is to increase wear resistance and reduce interfacial shear stresses and strains. This way, the material properties and performance of the surface layer can be significantly improved without altering the bulk materials [9–11].

Plasma nitriding is one of the most versatile nitriding techniques. It is a thermo-chemical process for diffusing nascent nitrogen onto the surface of steel or cast iron. It

increases the surface hardness and improves the fatigue strength of alloy steels causing minimum distortion or damage to bulk materials. However, the application of the nitriding technology for high (more than 13%) Cr-content plastic mould tool steels may impose limitations on their wide range of applicability. The reason for it can be the formation of the CrN precipitates during the high temperature (>450 °C) nitriding, accompanied by the simultaneous reduction in the free Cr content in the steel matrix leading to low corrosion resistance. This unfavourable structural change represents a key issue in the metal-mechanical industry, especially in the plastic injection moulding sector, where high alloyed tool steels are widely used. The general solution for this problem is the application of an anti-wear and anti-corrosion coating to enhance the tool's steel performance and durability.

Carbon-based coatings such as diamond-like carbon (DLC) have proven highly advantageous in tribological applications [12–14]. They consist of hydrogen and carbon atoms with $sp^2$ and $sp^3$ hybridisation. The classification of DLC coatings has been extensively reported in the literature [15–17]. The two main classes are hydrogen-free and hydrogenated films. The properties of DLC are largely dependent on the hydrogen content. An increasing amount of hydrogen makes DLCs less rigid and less stressed.

DLC coatings possess high hardness, chemical inertness, wear resistance, corrosion resistance, excellent adhesion to several substrate materials [18], self-lubricating capability, and low friction coefficient [19]. Applications of such coatings include automotive [20], aerospace industry, protective layer for cutting, forming tools, and biomedical implants [21–23].

These coatings have found wide applications in plastic injection moulds owing to high durability and influencing favourably specific technological characteristics, such as ejection force, surface lubrication, cavitation pressure, etc. The key requirement for a mould component coating is to effectively increase the productive service life of the components while reducing operational and maintenance costs. DLC coating on a plastic injection mould proved to show a non-sticking behaviour, with close to zero ejection forces [24]. A comprehensive overview of the beneficial effects of DLC coating in plastic injection mould tooling is provided by Delaney et al. [25]. Among the others, DLC significantly decreases the ejection force when polypropylene (PP)], polyethene terephthalate (PET) [26], polylactic acid (PLA), polystyrene (PS) [27,28], polyamide (PA), or polyoxymethylene (POM) [28] parts are removed from the tool. PVD-deposited DLC films—characteristic thickness of 0.5–5 μm—effectively reduced the cavity pressure for molten PS and PET plastics [29] and reduced the filling flow resistance of PET in thin-wall injection moulding.

Various technological solutions are available for depositing DLC coatings, e.g., magnetron sputtering (MS) [30,31], high power impulse magnetron sputtering (HiPIMS) [32,33]. The group of the plasma enhanced chemical vapour deposition (PECVD) techniques involves several types of processes, such as the radio-frequency plasma enhanced chemical vapor deposition (RFPECVD) [34,35] and the low-frequency plasma-enhanced chemical vapor deposition LFPECVD [36–38]. For industrial purposes, RFPECVD and LFPECVD are the most widely used methods due to their versatility in scale [39] and coating structure, e.g., for doped or multilayered DLC systems.

PECVD techniques are characterised by a low deposition temperature, allowing them to obtain a uniform layer structure and size and making them suitable for coating tool steel substrates of different treatment conditions. The efficiency of duplex treatment involving plasma nitriding was emphasised in decreasing the wear rate for AISI 4140 steel with MS DLC coating [40] and improving the wear and scratch resistance of the K340 cold work tool steel with PACVD DLC coating used as a deep drawing die in sheet metal forming [41]. Combined mechanical and thermochemical treatment—involving mechanical turning and burnishing combined with plasma nitriding—improved the tribological performance of multilayer PECVD DLC-coated Sverker 21 (AISI D2) steel and Vanadis 8 powder metallurgical (P/M) steel [42]. The importance of the effect of the surface finish and removal of the compound layer on friction and wear behaviour was emphasised in the case of MS DLC coated, quenched, and tempered and plasma nitrided AISI H11 hot work tool

steel [43]. Extended (up to 100%) tool life and improved surface quality of the plastic product were ensured by the DLC coating deposited on a plasma nitrided Vanadis 4 P/M austenitic stainless steel forming tool due to the outstanding anti-sticking properties and wear resistance of the DLC layer [44].

The major drawback suffered by these coatings on steel substrate is the occasionally poor adhesion that causes cracking under an external load. Doping with non-metal or metal is a suitable method to improve the thermal stability and enhance the tribological properties of the coating due to better adhesion [8,45]. The efficiency of a novel HiPIMS technique with positive pulses used for depositing DLC coating on K360, Vanadis 4, and Vancron steels was demonstrated in regard to higher wear resistance due to improved adherence of the coating [46]. The synergetic effect of low-pressure arc plasma-assisted nitriding (PAN) of M2 type steel and the PECVD technique on the adhesion of the DLC coatings was analysed by wear and scratch tests demonstrating the importance of the process parameters of both treatments on the adhesion and, consequently, the tribological behaviour of the coating [47].

The adhesion mechanism at the coating/substrate interface can be realised by mechanical interlocking and physical or chemical bonding or a combination [47]. While the effect of plasma nitriding on the improved adhesion properties of the DLC coatings is still being investigated, explanations by some researchers refer to the larger thickness of the intermixed layer at the interface and lower level of graphitisation—due to an increased bias voltage—during plasma nitriding [48,49]. These favourable effects may be further enhanced by an increased polarisation tension in the PECVD deposition process leading to better adhesion of the DLC coating to the M2 steel substrate [47].

When the coating is exposed to high loadings, plastic deformation begins in the vicinity of the coating/substrate interface. The applied coating can become damaged at this interface due to poor adhesion to the substrate or if the cohesive strength of the coating is poor. It was shown that a large plastic zone first develops in the substrate, followed by the initiation of plastic deformation in the coating. Thus, the substrate properties play a vital role in improving the load-bearing capacity of the coating-substrate system [50], which is evaluated quantitatively by the force required to remove the coating from the substrate, known as the critical force.

The instrumented scratch test is a simple, cost-effective, and widely used conventional method of determining the adhesive strength of the coatings to the interface. The result depends on several intrinsic factors, such as substrate mechanical properties, coating thickness, and interfacial bond strength, as well as extrinsic parameters such as scratching speed, applied normal load, type of indenter, tip radius, etc. The morphology of the scratch track and cracks initiating from the scratch groove with different orientations provide information on the controlling damage mechanisms such as cohesive or adhesive failure of coating; interfacial, tensile, or conformal cracking; brittle chipping, etc. The progressive load scratch tests provide the highest amount of information on the cooperative behaviour of the coated systems; therefore, such test results are extensively reported in the literature [51–54] for various combinations of the substrate/coating pairs.

Wear is defined as the irreversible material loss of interacting surfaces in relative motion. Physical and chemical elementary processes within the area of contact of a sliding pair leading, subsequently, to the change in material structure and geometry of the friction partners are known as wear mechanisms [55,56]. The most common ones are abrasion, adhesion erosion, fretting, and cavitation. Wear tests provide relevant qualitative and quantitative information on the wear resistance; the mechanism, severity, and mode of the damage; and materials performance among friction and wear type loadings that are useful during the design of the components to reduce material loss due to wearing [57]. A wear test configuration applied in engineering practice can be of great variety. It may realise, e.g., a point contact (ball on plate, ball on disk), a linear contact (block on plate), or a plane contact (block on ring, pin on disk) [58] to model the actual loading condition at the most adequate.

The substrate material in this work is X42Cr13 steel, which has a very similar chemical composition to that of the AISI 420-type steels. The literature research on plasma nitriding of X42Cr13 steel is found to a limited extent, while research on the AISI 420 martensitic stainless steel (MSS) is extensively reported [59,60]. The requirements of high surface hardness and simultaneous wear- and corrosion resistance is required for a wide range of applications. For low alloy steel, plasma nitriding is usually advantageous in improving corrosion resistance. In contrast, this treatment can have either positive or negative effects in the case of Fe-based stainless steels. The low-temperature plasma nitriding was suggested as a possible solution to enhance wear resistance without altering the excellent anti-corrosion properties that require a homogeneous distribution of the high Cr content. It can be assured by a low-temperature or short-duration nitriding operation, allowing the CrN precipitation in the nitrided layer to be avoided [61].

Another solution for enhancing wear resistance with simultaneous reservation of the corrosion resistance of high Cr steels is represented by the deposition of anti-wear ceramic coatings (such as DLC) on the previously nitrided steel substrate. In this regard, we report the first wear test results obtained on duplex-treated multilayer DLC/X42Cr13 tribosystems compared to those obtained on the control set of the un-nitrided and simple DLC-coated samples.

The primary objective of the current research work consists of comparing the tribological performance—in terms of the adhesion and wear behaviour—of two coated material systems, which have the same multilayer DLC top layer deposited onto the X42Cr13 plastic mould tool steel with two different surface conditions. We address the key issue of CrN precipitation in the tool steel at high-temperature nitriding, limiting the applicability of the duplex treatment in the case of this grade of tool steel. The reciprocating wear test configuration provides a good basis to elaborate on the harmful effects of CrN precipitation in the nitrided layer on the wear resistance of the coated system.

## 2. Materials and Methods

The tribological tests were carried out on the samples made of high-alloy plastic mould tool steel X42Cr13 (DIN) Nr. 1.2083. The chemical composition (in wt%) is C = 0.38–0.45, Si $\leq$ 1.0, Mn $\leq$ 1.0, P $\leq$ 0.030, S $\leq$ 0.030, Cr = 12.00–13.50, V~0.3, and the remainder is Fe. Disc-shaped samples with dimensions of $\Phi$30 mm $\times$ 10 mm were prepared.

The two types of the investigated coated systems are denoted by Sample (S) and Sample (D), where the identical DLC top layer is deposited on two different substrates. In the case of Sample (S), the substrate is bulk heat treated, that is, secondary-hardened steel, while in the case of Sample (D), this bulk heat treatment is followed by a subsequent plasma nitriding. Thus, Sample (S) is a simple DLC-coated system, while Sample (D) is a duplex-treated one. The technological parameters of the bulk heat treatment were as follows. Austenitisation was carried out at $T_{aust}$ = 1020 °C, for $t$ = 20 min, followed by cooling in a vacuum. The temperature and holding time of tempering was $T_{temp}$ = 580 °C and $t$ = 2 h in air.

The reason for this treatment is, on the one hand, to provide the best dimensional stability of the product, on the other hand, to increase the toughness, playing a vital role, for example, in low temperature and cryogenic applications, or dynamic loadings characteristic for tool materials. These treatment parameters represent a non-conventional treatment resulting in a lower substrate hardness than in most cases when this steel is used as tool steel.

The temperature of the plasma nitriding was 520 °C, with a holding time of 8 h, a voltage of 600 V, and a pressure of 2 mbar. The source of nitrogen was decomposed ammonia ($N_2$:$H_2$ = 1:3).

The DLC top layer deposited on Samples (S) and (D) by plasma enhanced chemical vapour deposition (PECVD) is a multilayer coating consisting of CrN + WC + aC:HW + aC:H layers. The coating is deposited using the equipment TSD 550 (from HEF Durferrit M&S SAS, Andrézieux-Bouthéon, France).

The process in the furnace starts with cleaning the samples by ion-etching using ionised Ar atoms to bombard the substrate for 60 min with 65 sccm argon (purity ~99.95%) under vacuum (pressure = $1 \times 10^{-3}$ Pa). This operation was executed using a microwave generator as an auxiliary plasma booster with 1000 W and 2.45 GHz frequency. The first step of the coating process is the deposition of the CrN and WC-Co underlayers.

The first sublayer was the CrN deposited at 150 °C for 30 min. The Cr target was sputtered using a power of 6.5 kW, and the gas flow of $N_2$ was controlled by PLC via light measurement of plasma. The second sublayer was the WC-Co interlayer created by magnetron sputtering. The target was a typical cemented carbide constituting of WC and 6% Co. For the deposition of the a-C:H top layer, acetylene gas with a flow rate of 200 sccm was utilised. The microwave generator served as an auxiliary plasma booster with 500 W and 2.45 GHz frequency.

The instrumented scratch tests were accomplished on a scratch tester (model SP15, Institute of Materials Science and Technology, the University of Miskolc, manufacturer: Sunplant, Miskolc, Hungary) [62]), conforming with the requirements of the ASTM C1624-22 standard [63]. During the applied progressive loading scratch test, a standard Rockwell C stylus (120° vertex angle and 0.2 mm radius) was drawn over the sample surface with normal force, increasing from an initial normal load of 2 N to a maximum normal load of 150 N until the coating was removed from the substrate. The gradient of the normal force was 10 N/mm, and the velocity of the table supporting the sample was 5 mm/min. The total scratch length was 15 mm. Tests were made on two pieces of the simple- and duplex-coated batches producing three scratches per sample. The diamond stylus was cleaned by manual polishing after each scratching using a polish cloth (grade Struers MD-NAP) soaked with acetone. Then, the tool was sonicated in acetone for 3 min to remove the scratch-test-originated residuals. The integrity of the stylus was checked by OM after each test. The critical loads were obtained from the scratch diagrams registered by the test machine, while subcritical loads and damage characteristics were evaluated by optical microscopy based on the ASTM C1624-22 standard.

The wear tests were carried out in the laboratory of Rtec Instruments, Switzerland using a multi-functional tribometer (type MFT-5000, from Rtec Instruments, Yverdon-les-Bains, Switzerland) equipped with a load cell (200 N) and an imaging system using a confocal microscope. Two samples of simple- and duplex-coated types were tested, carrying out two measurements on one piece. To model the fretting-type loading, generally occurring during the operation of forming tools, a reciprocating wear test configuration was chosen to characterise the wear behaviour of the samples. This type of test regime is widely used for tool coatings. Wear tests were accomplished with a normal load of 25 N, a reciprocating frequency of 5 Hz, a sliding distance of 10 mm, and a cycle number of 7900 at room temperature, in dry sliding conditions, at a relative humidity of RH = 50%. The material of the frictional counterpart was an $Al_2O_3$ ball of 6 mm in diameter, using a new ball for each measurement.

The coating hardness was characterised by the composite hardness of the coated systems using a standard micro-Vickers method with $F$ = 0.5 N loading force. This test was performed with $F$ = 0.25 N normal load to define the hardness profile in the nitrided layer to obtain the nitrided layer depth (Mitutoyo HVK-H1 hardness tester from Mitutoyo Corp., Kanagawa, Japan). Two samples from each group (simple and duplex-treated) were tested by performing ten valid measurements—having no cracks from the corners—by sample. Three profiles on each of the two duplex-treated samples were determined.

To evaluate the thickness of coatings, the ball cratering method was used (Compact CAT²c Calotester from Anton Paar GmbH, Graz, Austria) by creating three craters on two samples of both the simple- and the duplex-coated groups [64,65].

The structure of the coatings and the substrate were examined by scanning electron microscopy (Zeiss Evo MA10, form Zeiss, Jena, Germany) equipped with an energy-dispersive X-ray spectroscopy. Optical microscopy (Zeiss, Axio Observer D1m from Zeiss,

Jena, Germany) using the Axio Vision imaging software (Zeiss, Jena, Germany) was applied for the morphological analysis of the scratch grooves.

## 3. Results and Discussion

### 3.1. Coating Thickness and Structure

The architecture of the multilayer coatings is shown in Figure 1. They are built up with a CrN bottom layer (thickness of 0.7 µm for both samples), an a-C:HW intermediate layer (0.8 and 0.9 µm for the simple and duplex coating, respectively), and an a-CH top layer (thickness of 1.9 and 2 µm, respectively). The thickness of coatings, defined by the ball-cratering method on the simple and duplex coatings, were $3.4 \pm 0.07$ and $3.6 \pm 0.06$ µm, respectively.

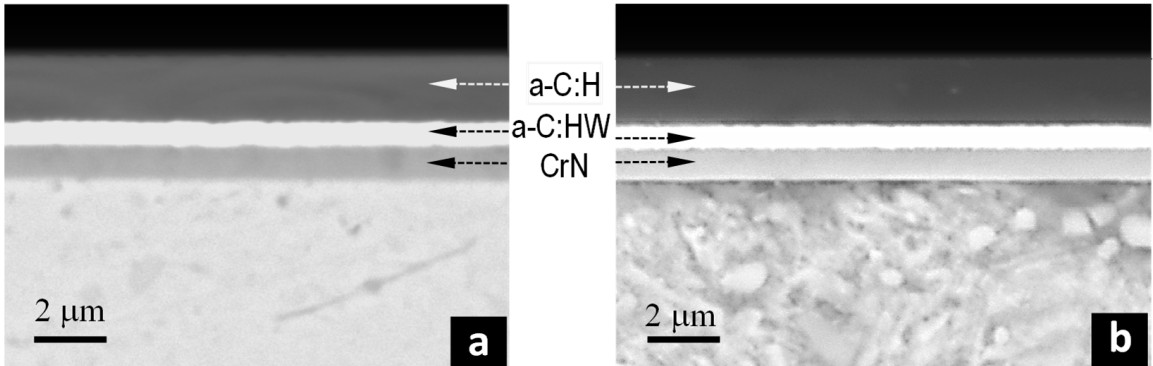

**Figure 1.** The architecture of the DLC coating on the simple-coated (**a**) and duplex-treated (**b**) samples taken by optical microscopy.

### 3.2. Microhardness

The average depth of the nitrided layer is 0.14 mm for the duplex-treated sample, which was defined based on the hardness profile below the surface (Figure 2).

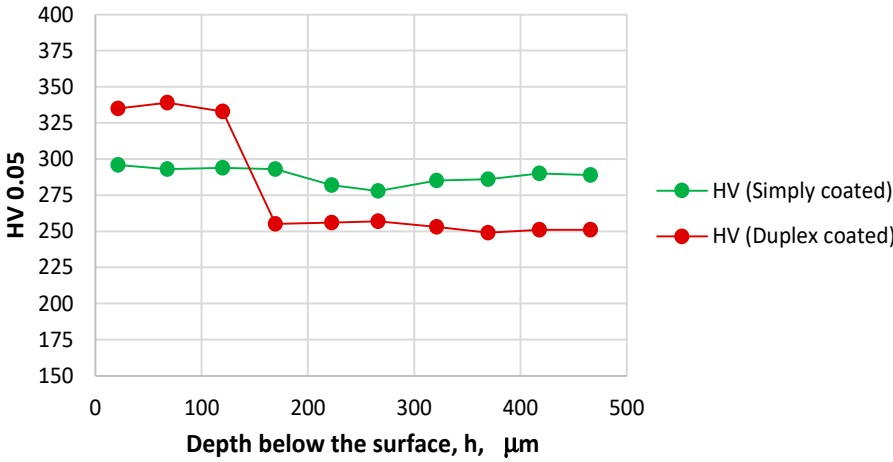

**Figure 2.** Microhardness profile of the simple-coated and duplex-treated sample.

The coating characteristics are summarised in Table 1. It is seen that DLC coating caused no changes in the substrate hardness (see HV of the substrate surface and substrate core for Sample (S). At the same time, plasma nitriding increased the surface hardness while decreasing the core hardness of the secondary hardened substrate in the case of Sample (D).

**Table 1.** Sample nomination and characteristics of the multilayered DLC coating.

| Sample | | Substrate Condition | Coating | | | | |
|---|---|---|---|---|---|---|---|
| | | | Thickness, μm | | Hardness, $HV_{0.05}$ | | |
| Nomi-Nation | Type | | DLC Coating | Nitrided Layer | DLC Coating * | Substrate Surface | Substrate Core |
| S | simple-coated | un-nitrided | 3.4 | – | 1386 | 295 | 290 |
| D | duplex-treated | nitrided | 3.6 | 140 | 2077 | 335 | 250 |

\* The coating hardness is characterised by the composite hardness of the coated system.

### 3.3. Microstructure Characterisation

A magnified SEM cross-sectional image of the nitrided layer produced at 520 °C for 8 h holding time is shown in Figure 3. The image was captured using the backscattered electron (BSE) mode to visualise the microstructural features and compositional variations in the nitrided surface. Elemental analysis by EDX in three characteristic points was made. Chemical composition at Spot 1, close to the spherical precipitate, and at Spot 3, which was taken in the middle of the grain, at a somewhat higher depth (~3 μm) below the surface, was almost identical. The increased Cr and N content suggest the presence of nitrides, which appear as dark grey areas around the spherical precipitation and at spot 3 in the BSD imaging mode. Spot 2, taken along the grain boundary (GB) region, shows higher Cr, N content compared to Spots 1 and 3, which give strong evidence of the path of CrN precipitation along the GB.

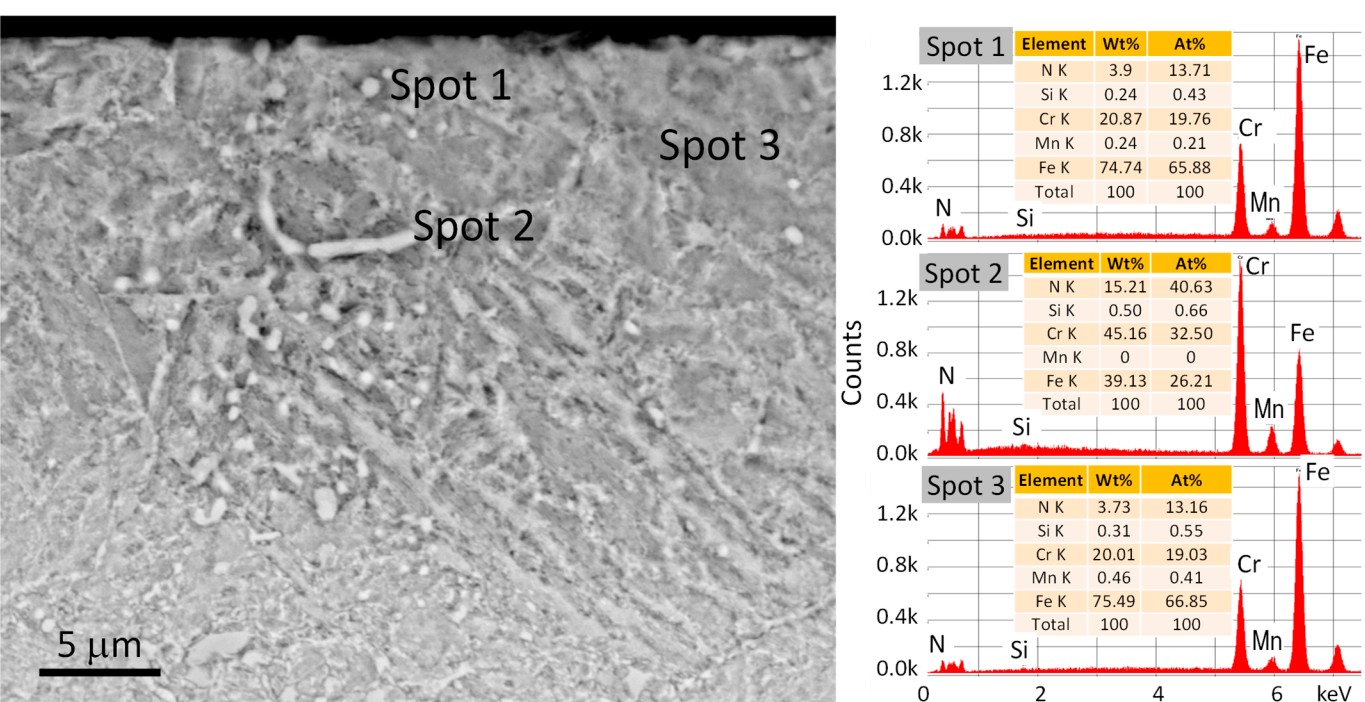

**Figure 3.** The SEM image of the nitrided zone of the (S) sample with the EDX spectra taken in three characteristic locations of the layer: Spot 1—close to a precipitate in the near-surface region inside the grain, Spot 2—a grain boundary precipitation, Spot 3—the grain volume at a slightly deeper region below the surface.

The GBs are clearly seen due to the heterogeneous deformation among the grains, resulting from the compressive residual stresses generated by the diffusion of nitrogen from the surface to the core, accompanied by simultaneous nitride precipitation [66].

*3.4. Scratch Resistance*

The scratch diagrams for the simple-coated (S) and duplex-treated (D) samples recorded during the progressive loading scratch tests are shown in Figure 4. which provides a comparative analysis of the friction coefficient ($\mu$) vs. scratch length for the two coated systems with the differently treated substrate. In the first approximation, it is seen that the critical ($L_{c3}$) and subcritical ($L_{c1}$ and $L_{c2}$) loading forces are higher for the duplex-treated sample.

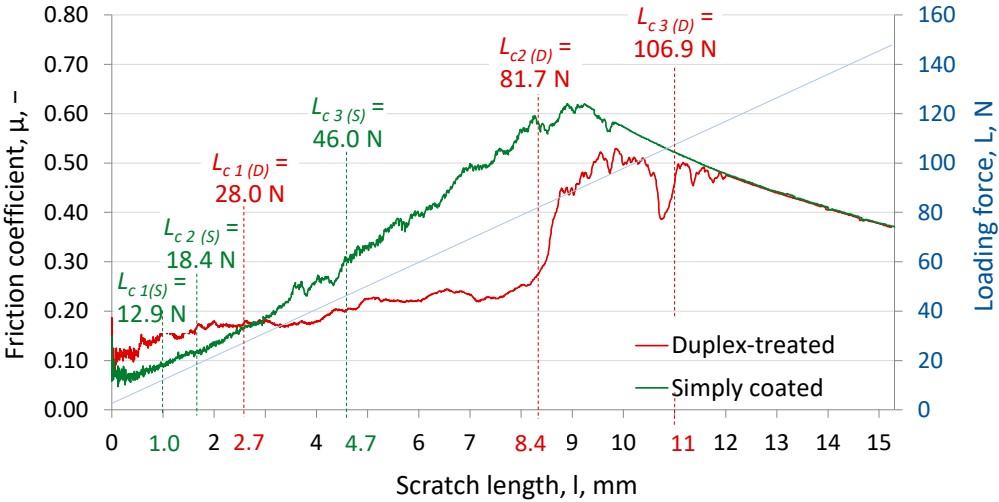

**Figure 4.** Scratch test diagrams for the simple-coated and duplex-treated samples.

These load levels indicate the initiation of specific characteristic damage processes identified using the optical micrographs shown in Figure 5.

The friction coefficient curve for the simple-coated sample (Figure 4) illustrates that the initial micro-cracking (Figure 5a) takes place at 1.0 mm at a loading force of $L_{c1\,(S)}$ = 12.9 N, followed by buckling spallation (Figure 5b) observed at the scratch length of 1.7 mm and a normal load of $L_{c2\,(S)}$ = 18.4 N, where $\mu$ = 0.110. At the critical force of $L_{c3\,(S)}$ = 46.0 N, when the friction coefficient is $\mu$ = 0.217, the coating becomes detached from the substrate, making the highly deformed, periodically torn substrate visible (Figure 5c).

Once the coating is removed, there are traces of detached coating material beyond the groove. It is indicative of gross spallation (Figure 5c), a characteristic failure mechanism of coatings with low adhesion strength [63]. The reason behind the increasing value of $\mu$ is the presence of the small coating's particles removed already by the subcritical loads, which cling in the vicinity of the stylus, creating additional resistance to the motion of the stylus and thereby increasing the $\mu$ value.

Regarding the friction coefficient curve obtained for the duplex-treated sample, the critical force at which the coating delamination begins is significantly higher, that is, $L_{c3\,(D)}$ = 106.9 N. The $L_{c1\,(D)}$ and $L_{c2\,(D)}$ subcritical forces are 28.0 and 81.7 N, respectively, which initiate microcracking (Figure 5d) and arc tensile cracking (Figure 5e) of the coating. Beyond the $L_{c3\,(D)}$ loading, there are traces of rounded regions of DLC coating that are removed laterally from the edges of the scratch groove. It is known as chipping [63], which was observed in the case of the duplex sample. It suggests a better scratch resistance of the coating and strong adherence [63] to the substrate. Thus, it can be concluded that the duplex sample possesses a higher critical load, higher scratch resistance accompanied by different damage mechanisms, and stronger adherence to the substrate compared to the simple-coated sample. The improvement is 117, 344, and 132% in terms of the $L_{c1}$, $L_{c2}$, and $L_{c3}$ subcritical and critical loads, respectively. The authors of the current work experienced a similar improvement in a recent round robin test [67], as well as found by others [40,53] for hydrogenated DLC coating if duplex treatment was applied.

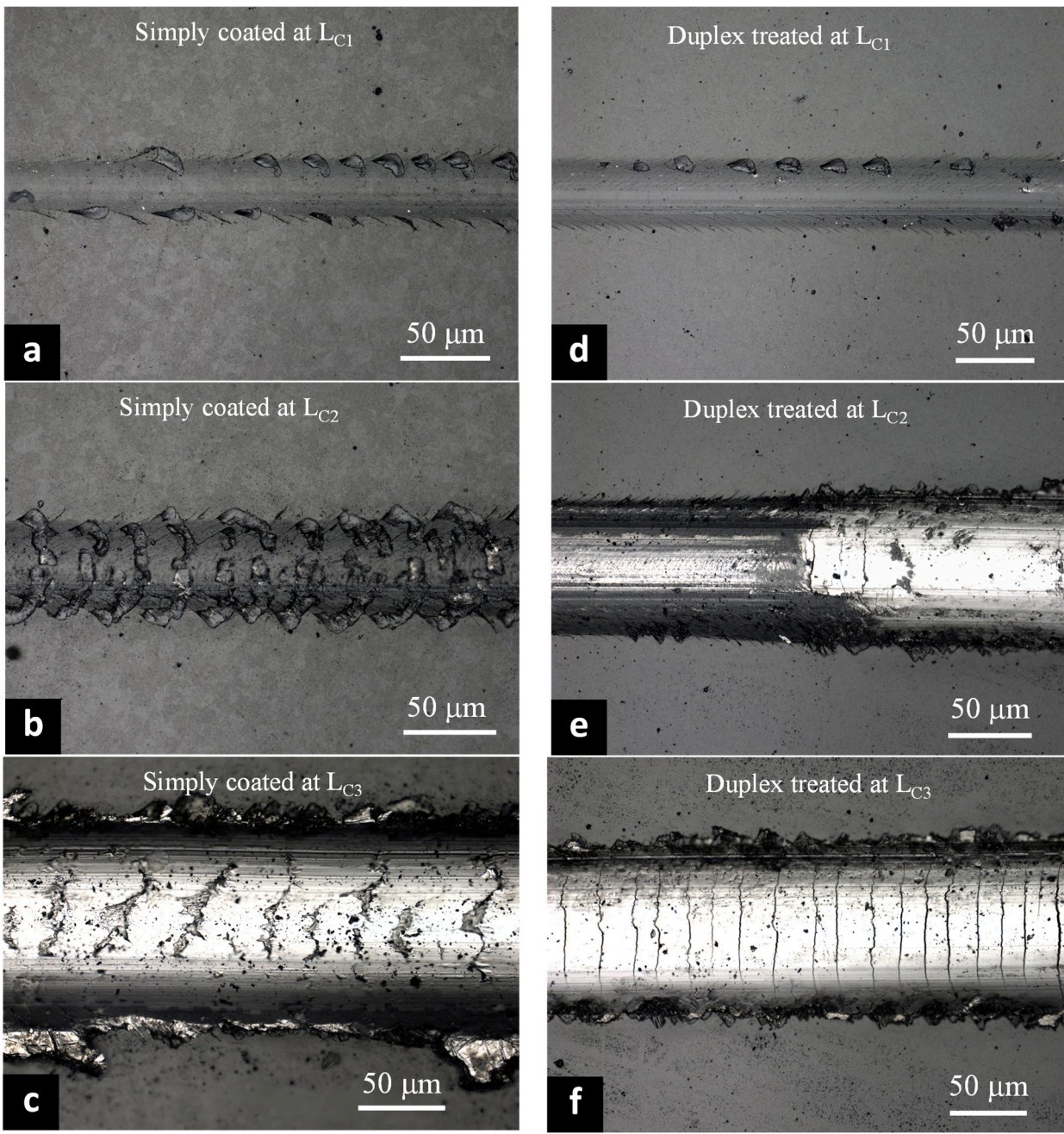

**Figure 5.** Optical micrographs illustrating the characteristic damage mechanisms that occurred at the subcritical and critical loadings in the simple DLC coated system (**a–c**) and the duplex-coated DLC system (**d–f**).

*3.5. Wear Resistance of the Simple and Duplex-Treated Coatings*

The friction coefficient curves for Sample (S) and (D), obtained during the reciprocating wear test, are shown in Figure 6. Surprisingly, the steady state friction coefficient is twice as high for the duplex-treated sample ($\mu_{(S)}$ = 0.049) than the simple-coated one ($\mu_{(D)}$ = 0.020). At the beginning of the test, that is, from 162 to 300 s of running, an abrupt increase in the friction coefficient—from 0.06 to 0.22—is also seen in the case of the duplex coating.

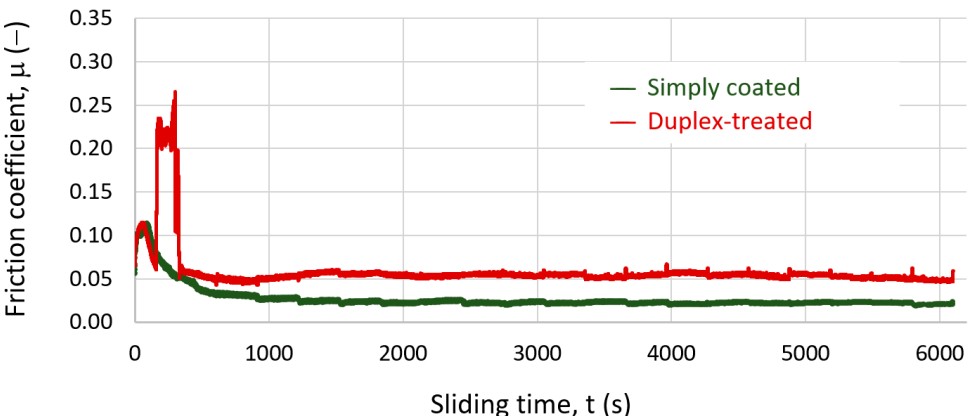

**Figure 6.** Friction coefficient vs. time curves obtained during the wear testing of the simple-coated and duplex-treated samples.

The 2D and 3D profilometry images for the disks (Figure 7) and the balls (Figure 8) provide qualitative and quantitative analysis of the wear damage produced on the two different coating systems. The worn track is significantly thinner for the simple coating showing negligible debris outside the worn path. In contrast, a large amount of debris decorates the two sides of the wear track in the case of the duplex coating.

Regarding the quantitative characteristics, the worn width values are $w_S$ = 165 μm and $w_D$ = 245 μm, and the worn depth values are $d_S$ = 0.44 μm and $d_D$ = 0.86 μm on the simple-coated and duplex-coated samples, respectively. Similar differences in the worn scar diameters, that is, $x_S$ = 155 μm and $x_D$ = 247 μm for the simple and duplex coating, were measured on the worn alumina balls (Figure 8). These results represent a 48%, 93%, and 59% increase in wear width, wear depth, and ball scar diameter, respectively. Thus, we can conclude that during reciprocating wearing, a significantly higher amount of material loss occurred in the case of the duplex coating, that is, the wear resistance decreased considerably. Nevertheless, the integrity of the DLC coatings was kept in both coating systems in terms of the wear depth remaining below the thickness of the a-:CH top layer.

In analysing the reason for this unexpected tribological response of the duplex-treated sample, the following considerations can be made. The $F$ = 25 N normal load caused a reversal motion of the sample and the ball in every test cycle of the reciprocating test. How can this cause more significant damage to the duplex coating? The most probable answer is given by the more enhanced Bauschinger effect (BE) in the secondary hardened and nitrided substrate material. Ellermann and Scholtz studied the BE in different heat treatment conditions of 42CrMo4 steel and found that the larger amount of carbide precipitates causes larger BE [68]. Queyreau and Devincre made dislocation dynamics simulations to analyse the Bauschinger effect in precipitation-strengthened materials. They established that the most effective contribution to BE is given by the Orowan-looping around the II. phase particles, interacting with the mobile dislocations. Thus, an increasing volume fraction of unshearable particles contributes extensively to the larger BE. In addition to the volume fraction, the density, size, and shear strength of these precipitates also have a definite role in the BE [69]. Kostryzhev et al. demonstrated that a higher amount of larger precipitates in differently treated C-Nb and C-Nb-V steel plates are responsible for a higher BE stress parameter [70]. The BE is strongly related to the residual stresses [68,71], that is, it may be enhanced by the large tensile stresses usually forming around the precipitates [72].

These observations support the hypothesis that the underlying duplex-treated substrate material below the hard DLC coating was softened to a greater extent during the reversed shear plastic loading caused by the friction force. It can be explained by the higher amount of precipitates present in the nitrided layer due to the high-temperature nitriding applied.

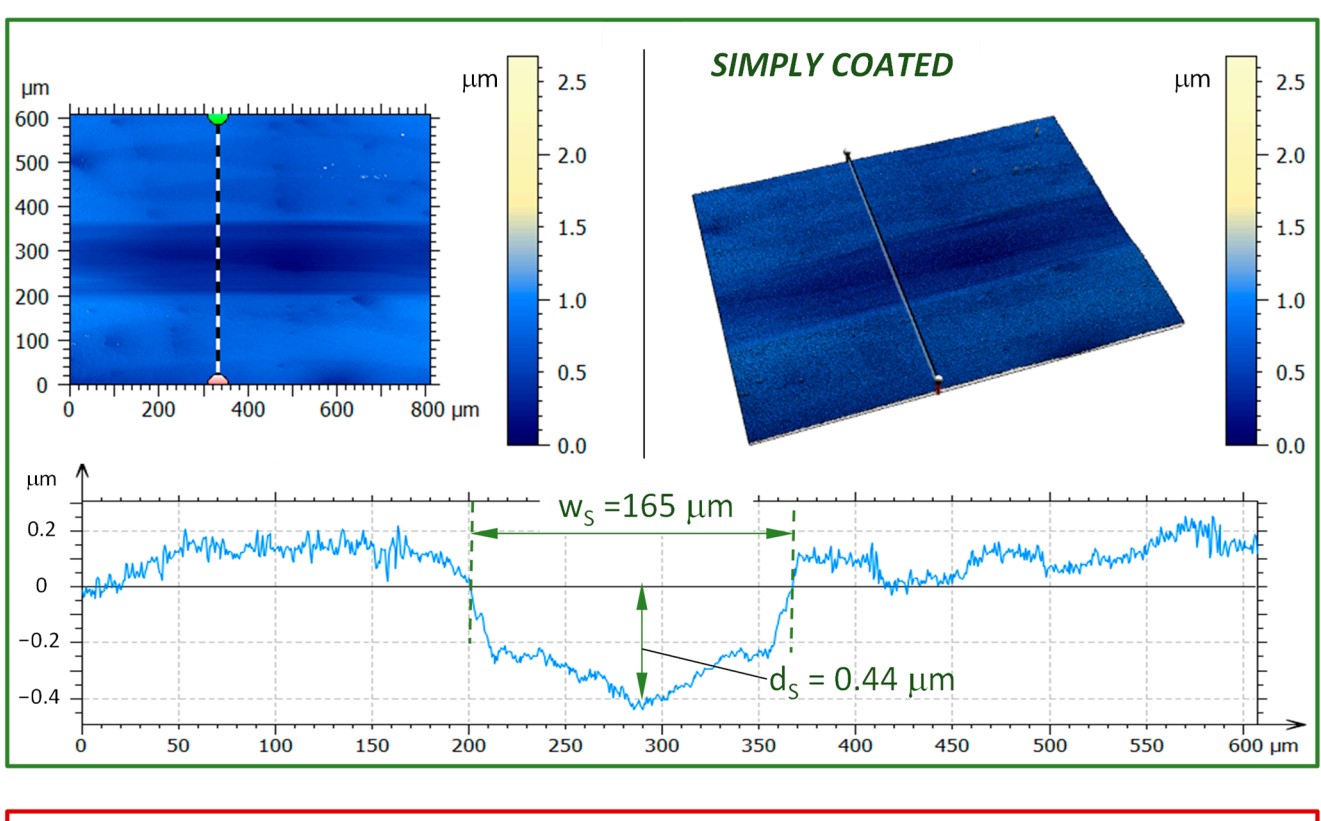

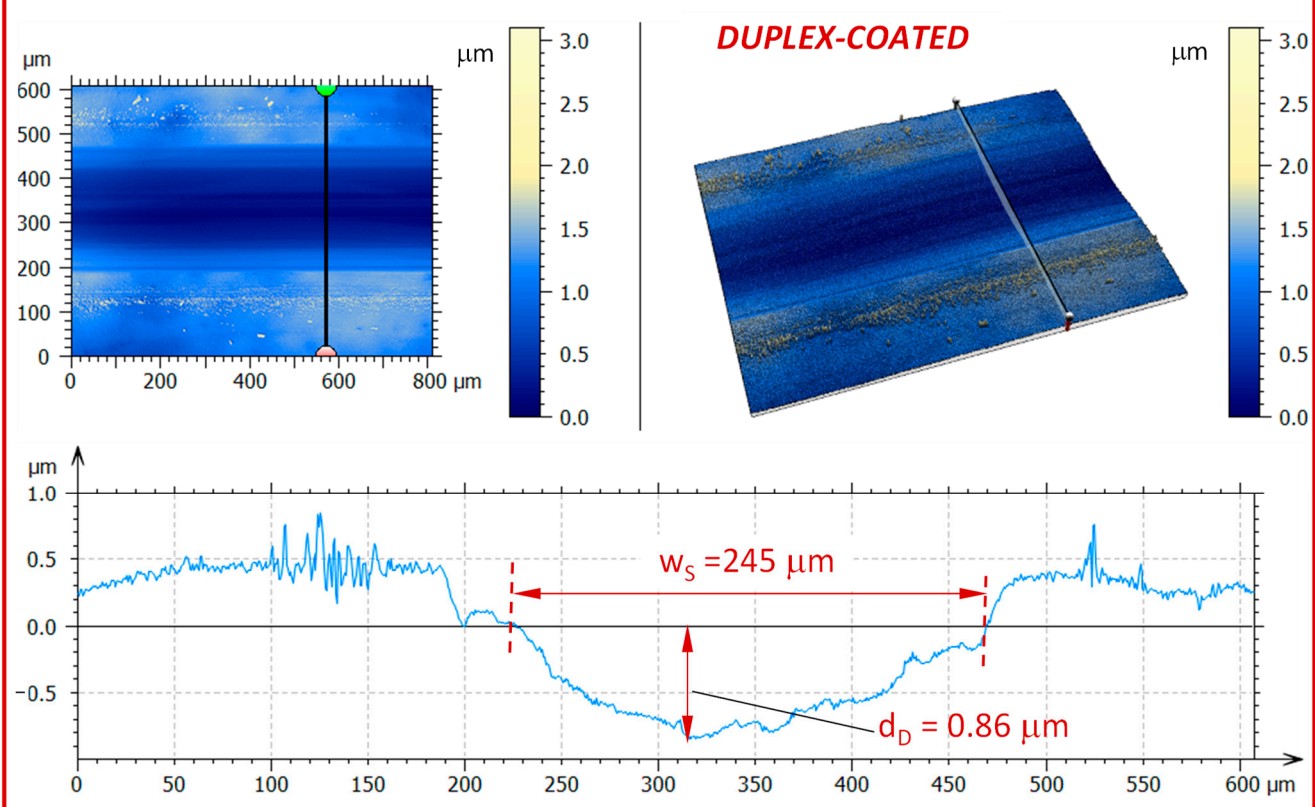

**Figure 7.** 2D and 3D profile of the wear track on the simple-coated (**up**) and the duplex-coated (**down**) discs.

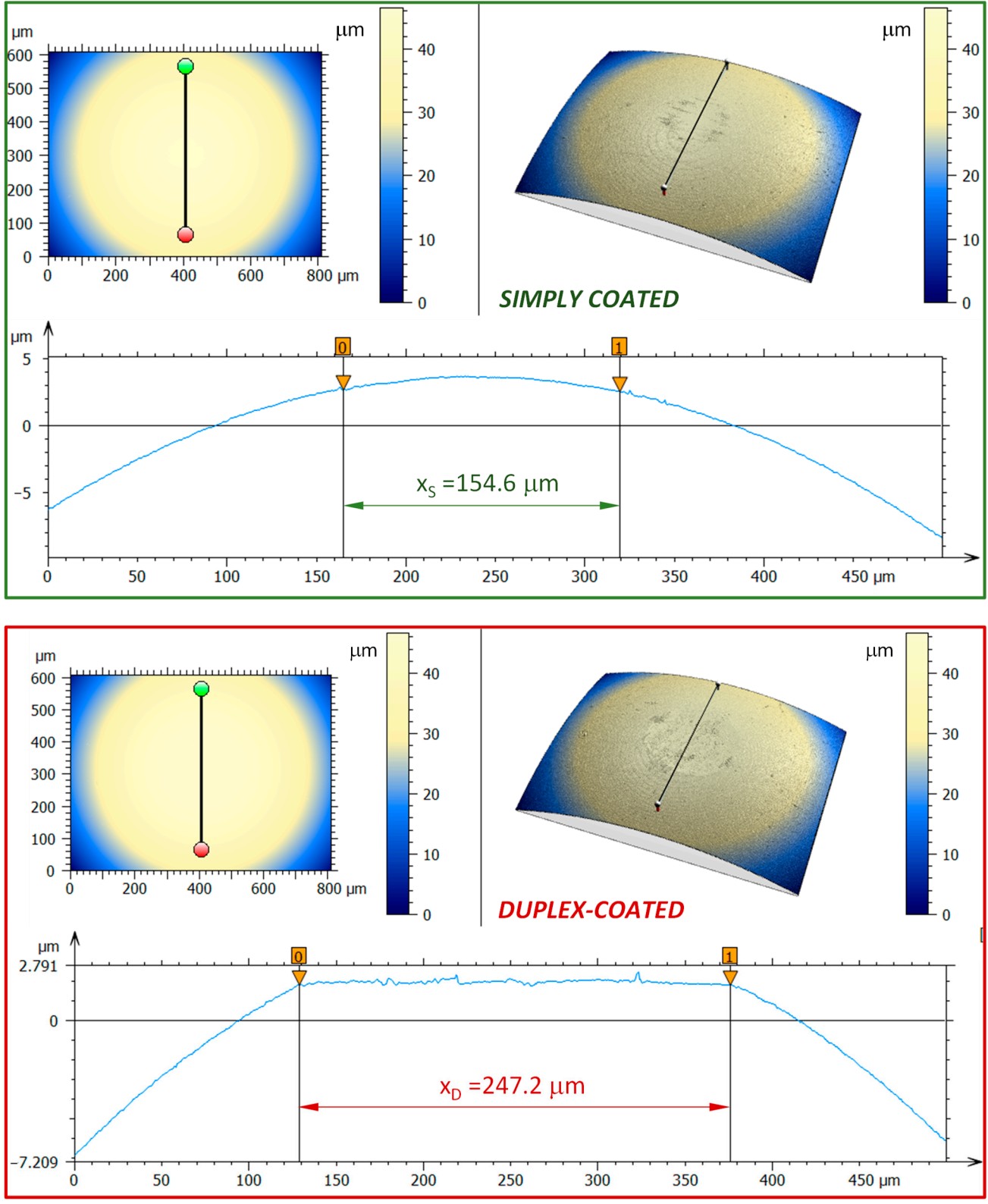

**Figure 8.** 2D and 3D profile of the wear track of the ball counterpart of the simple-coated (**up**) and the duplex-coated (**down**) discs. Consider the different scales along the vertical axes of the profile curves.

Similarly, an unexpected decrease in the hardness below the worn surface was reported by Mussa et al. during the reciprocating wear test of the case-hardened martensitic

22NiCrMo12–F steel. At the same time, they observed strain hardening for the same material in unidirectional loading of the ball-on-disc test [73].

It should also be noted that the nitrided layer, providing generally better support for hard coatings, was in this case, relatively thin due to the high Cr content of the steel preventing efficient N diffusion into the surface (see Figure 2). Thus, a compositionally and microstructurally heterogeneous subsurface layer structure was developed, unfavourable regarding the cyclic fatigue during the wear test. As a result, the hard coating was easily broken on the softened substrate producing hard-wear debris particles, enhancing and accelerating the damage process and leading to severe wear due to a three-body abrasive wear mechanism.

The main reason for the failure of the DLC coating altogether is the softening of the substrate material, resulting in an intensive plastic deformation in the metallic substrate during both the fro- and the reversal motion of the ball. The intensive shear deformation of the soft substrate towards the two sides of the groove may contribute to the high tensile stresses in the DLC coating. The accumulation of the plastic deformation during the subsequent passages of the loading reveals itself in the ploughing mechanism, traces of which are visible on the worn surface, as shown by the 3D OM image of the wear track of the duplex-coated specimen in Figure 7.

The load condition is further complicated by residual stresses around the precipitates in the high-Cr nitrided steel, which can initiate intensive cracking in the base material. Their stress-concentration effect can also contribute to the increase in tensile stresses arising in the coating.

A considerable contribution to the amount of abrasive particles may be given by the broken particles from the alumina ball, the amount of which was significantly higher while wearing the duplex-coated system.

The magnified image of the initial region of the friction coefficient vs. time diagram reveals that the two curves of the simple-coated and duplex system moved close to each other for approximately 160 s (Figure 9). The friction coefficient of the duplex-treated sample decreased slightly faster than that of the single-treated sample after the local maximum associated with the initial stick-slip behaviour. At 160 s, an abrupt increase in the friction coefficient indicated the premature failure of the duplex-treated coating. The very hard ceramic coating particles caused an increase in the friction coefficient to about four times greater than the friction coefficient, exceeding the 0.2 value. The detached particles of the extremely hard coating caused a three-body abrasive wear until they became chopped due to the repeated rubbing of the mating surfaces. After approximately 300 s, the coefficient of friction abruptly fell and showed a similar character over time as the curve of the simple-coated sample; however, its value always moved above it.

If neglecting the deviations in the initial stage, the friction coefficient curves typically show a decreasing trend with time for both coating systems, which indicates the tribochemical nature of the wear of the DLC coatings.

The explanation for the fact that the friction coefficient is higher, approximately double for the duplex-coated specimen compared to the simple-coated one, is the substantial softening of the substrate due to the more considerable plastic deformation in the reversed direction. It contributes to a much more intensive crack formation and propagation, consequently, a more significant amount of debris formation in the DLC top layer, which accompanies the entire wear process, as seen in Figure 6.

The presented tribological investigations show that the applied duplex treatment—namely, the high-temperature plasma nitriding of the steel substrate before the coating deposition—is inefficient in offering higher wear resistance during reciprocating sliding compared to that of the simple-coated samples.

It should be noted that ball-on-disk wear tests—realising unidirectional circumferential loading conditions—on the same batch of simple and duplex-coated samples prevailed better wear resistance of the duplex-coated tribosystems [74]. The similar wear behaviour

of cemented 22NiCrMo12–F substrate in ball-on-disc, as well as reciprocating wear test configuration, was observed by the authors of the work [73].

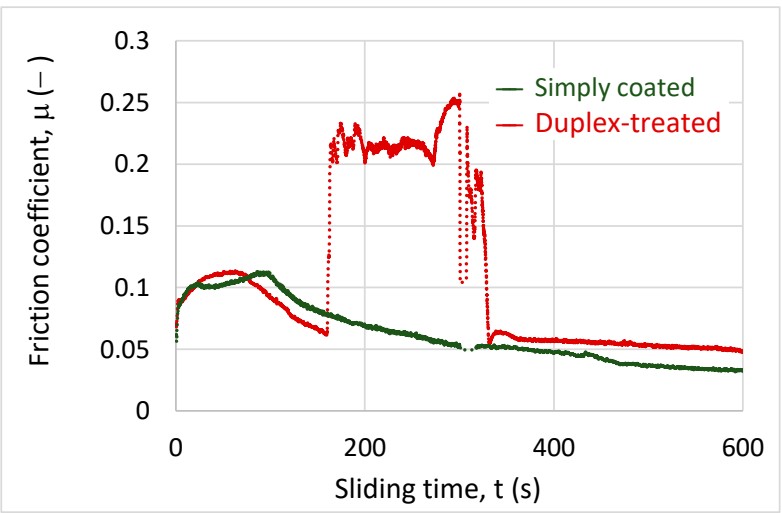

**Figure 9.** The initial part of the friction coefficient vs. time diagram illustrating the different wear behaviour of the DLC coating on the simple- (S) and duplex-treated substrate.

At the same time, we could demonstrate that during the scratch tests, where only unidirectional loading occurs, plastic softening of the substrate in the duplex-treated specimens did not happen, and the favourable effect of the combined bulk and surface treatments prevailed.

These experiences draw attention to the fact that the problem of high-temperature nitriding of steels with a high Cr content is not yet solved. The optimal technological parameters must be developed for the given steel grade based on further experimental work. In addition, our hypothesis on the modified BE behaviour of the duplex-coated system, in reciprocal and unidirectional wear test conditions, has to be supported by further, purposefully designed comparative wear tests supplemented with detailed microstructural and phase analysis of the worn surfaces.

## 4. Summary

The current study evaluates the adhesion and wear resistance of two hard coating systems represented by the identical multilayer DLC top coatings deposited on nitrided and un-nitrided steel substrates. The substrate material was a secondary hardened X42Cr13 tool steel involving tempering to high toughness and dimensional stability. Based on the results of the tribological tests performed, the following conclusions can be drawn.

The applied duplex treatment benefits the tested DLC coating system in terms of scratch resistance. The $L_{c3}$ critical load was improved more than 130% and the $L_{c1}$ and $L_{c3}$ subcritical loads were doubled and tripled, respectively.

In contrast, the applied surface treatment cannot be efficiently used for the tested combination of substrate material/heat treatment/multilayer coating in respect of the reciprocating wear test accomplished with the given test circumstances.

Despite the better adhesion strength of the coating for the duplex sample, shown by the progressive loading scratch test, the duplex-coated system offers poor wear resistance in the reciprocating wear test configuration compared to the simple-coated ones. The reason behind the unfavourable behaviour is the intense CrN precipitation during the nitriding process and particle coarsening, which on the one hand, induces an intensive plastic softening of the substrate due to the enhanced Bauschinger effect, a characteristic of the reversed motion during reciprocating sliding. On the other hand, precipitates in the nitrided steel subsurface region may induce high residual tensile stresses, contributing to the tensile stresses developing in the DLC coating due to the shear deformation of the

softened steel substrate. As a result, a more enhanced cracking and failure of the DLC coating occur, accompanied by a more intensive debris formation in the reciprocating wearing of the duplex-treated coating.

These results raise the need for a more profound overview of these novel findings, during which both purposefully designed new comparative wear tests, deeper microstructural and phase analyses of the worn substrate, compositional and morphological analysis of the debris particles, as well as estimation of the developed stress state by simulation software (e.g., ABAQUS, 3D-FEM) have to be taken into account as tools of the solution.

The direction of future work is to analyse the effect of the different loading conditions embodied by the other test methods on the wear behaviour of the duplex-treated coating with particular attention to the mentioned features. In addition, the influence of the core hardness of the substrate material—plastic mould tool steel—controlled by the tempering temperature during bulk heat treatment should be optimised for better wear performance of the coated systems.

**Author Contributions:** Conceptualisation, methodology, M.B.M.; test execution, data processing, S.A.S.; writing—original draft preparation, S.A.S.; writing—review and editing, M.B.M.; investigation, data evaluation and analysis, S.A.S. and M.B.M.; visualisation: S.A.S. and M.B.M.; supervision: M.B.M. All authors have read and agreed to the published version of the manuscript.

**Funding:** This research received no external funding.

**Institutional Review Board Statement:** Not applicable.

**Informed Consent Statement:** Not applicable.

**Data Availability Statement:** The data presented in this study are available on request from the corresponding author.

**Acknowledgments:** The authors thank the support of the Stipendium Hungaricum scholarship by the Tempus foundation, Hungary. The authors express their gratitude to Attila Széll and Gyula Juhász at T.S. Hungary; Andrea Szilágyiné Biró at Oerlikon Balzers GmbH Kft., Székesfehérvár, Hungary; and Norbert Luczai at Voestalpine High-Performance Metals Hungary Kft. for their assistance in heat treatment, coating, and valuable professional consultations. We also thank the colleagues at the University of Miskolc, namely, Csaba Felhő for the profilometry analyses, Árpád Kovács for the SEM and EDX measurements, and Gregory Favaro and Vishal K. Khosla at Rtec Instruments, Switzerland for the reciprocating wear tests. The authors thank the Institute of Materials Science and Technology of UM for the financial support covering the costs of the mechanical, tribological, and microstructural testing.

**Conflicts of Interest:** The authors declare no conflict of interest.

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
