# Peer review of "Comparative Study on the Scratch and Wear Resistance of Diamond-like Carbon (DLC) Coatings Deposited on X42Cr13 Steel of Different Surface Conditions"

_ceramics, doi:10.3390/ceramics5040086_

Round 1

Reviewer 1 Report

The paper presents experimental work on mechanical testing of two plastic mould steel samples treated for increased tribological properties. The whole report contains data related to two samples only and no important conclusions can therefore be drawn. A parametric study to observe the influence of a certain parameter on mechanical properties of a series of samples would have given a more useful result.

Some aspects that could improve the text are proposed:

Introduction:

The paragraph on types of carbon-based coatings is redundant.

Literature survey on the state-of-the-art on deposition of DLC on moulds and generally on other steels would add more value to this section.

Materials and Methods:

This section contains some data which should be part of the Results section (Figures 1-3 and associated text, tables).

Results

Figure 4 should be rearranged so that all SEM images are positioned close to each other for easier comparison. Images and graph should also be split into two separate figures.

Figure 5. Explanations concerning the spikes in the spectra should be given.

Author Response

Thank you for your detailed comments and suggestions. We have attached a detailed response letter as a pdf document. 

Reviewer 2 Report

1. (Line 181 and Fig.2) The hardness loading force data is inconsistant. Line 181 states the force is 5N, while the data in Fig.2 is 0.05kg (0.5N).

2. Table 1 is presented twice in page 5.

3. (Line 148-150) If the heat treatment processes were carried out under vacuum or protected atmosphere environment, it is suggested to add notes in this paragraph.   

4. More descriptions could be added regarding to the tailored scratch tester, or reference could be given. For example, the material and surface condition of the stylus may have great influence towards the scratch result. Also, the stylus should be polished or replaced after one trial of scratch test the minimize the accumulated surface wear on the stylus.

5. It is suggested to change the unit of 2D wear track profile of simply coated specimen (Fig. 6) from nanometer to micron, so that a better comparison with the wear track of duplex coated specimen could be made.

6. A conclusion is made that the duplex coating shows better scratch resistance capability and weaker performance in the applied wear type of loading. More discussions should be made in order to clarify this phenomenon, for the hardness of duplex coated layer is higher, better adhesion strength is proposed, while larger wear depth is obtained.

What is the dominating factor of this result? How does the reciprocating move of the Al2O3 ball influence the wear behavior differently comparing with scratch test?

Does the softer substrate influence the deformation amount (lump at both sides of the groove) during wear test, so that the DLC coating bears higher tensile strength and thus fractures? 

Author Response

(The authors gave the same response as above.)

Round 2

Reviewer 2 Report

The proposed questions and suggestions were dealt with, and the contents related were revised with sufficient explanations.